

# Impact of three weeks of integrative neuromuscular training on the athletic performance of elite female boxers

Zhen Niu[1], Zijing Huang[2], Gan Zhao[2] and Chao Chen[3]

[1] School of Physical Education, Shanghai University of Sport, Shanghai, China
[2] School of Athletic Performance, Shanghai University of Sport, Shanghai, China
[3] Dalian University College of Physical Education, Dalian University, Da Lian, Liao Ning, China

## ABSTRACT

**Objectives.** To investigate the effects of integrative neuromuscular training (INT) on the athletic performance of elite female boxers.

**Methods.** A before-and-after controlled experiment was conducted on 37 elite Chinese female boxers (Age: $26.00 \pm 3.11$ years). All included athletes have competed at the international level. The INT intervention was administered 11 times per week for 3 weeks. This training includes strength training, explosive training, core stability, agility exercises, high intensity intervals and sprint intervals. Basic physical fitness tests, including the deep squat and bench press one-repetition maximum (1RM), vertical long jump, 30 m sprint run, 400 m run, 3,000 m run, 1-minute hexagonal jump, and 3-minute double shake; as well as specialized striking ability tests, including single-punch striking and 10-second, 30-second, and 3-minute continuous punching, were conducted before and after the intervention.

**Results.** Compared with pre-intervention baseline data, significant differences were found in the athletes' post-intervention baseline physical fitness, including squat and beach press (1RM), vertical jump, 30 m sprint run, 400 m run, 3,000 m run, 1-minute hexagonal jump, and 3-minute double shake ($p < 0.05$). Additionally, 10-second, 30-second, and 3-minute continuous punching were significantly different compared with pre-intervention ($p < 0.05$). However, no significant differences were found in single punch power ($p > 0.05$).

**Conclusion.** The 3-week INT can significantly improve the maximum strength, vertical explosive power, linear acceleration, agility, and continuous punching abilities of Chinese elite female boxers. The use of INT in physical training may enhance their athletic performance.

## INTRODUCTION

Boxing has been called the "Noble Art" and is one of the oldest combat sports in human culture (*Chaabéne et al., 2015*). Amateur boxing competition consists of $3 \times 3$ rounds with 1 min of rest for each round; the player needs to hit the effective part of the opponent's body to score points; apart from the short duration of the program, the intensity of the

Corresponding author
Chao Chen, taishanchenchao@126.com, chenchao2019@sus.edu.cn

activity is high; to win the competition, boxers should possess a high level of technical and tactical literacy and good physical fitness and physiological health (*Davis, Wittekind & Beneke, 2013*; Chaabène et al., 2015). The efficacy of a boxer's specific movement is significantly influenced by the velocity and synchronization of the motion, the precision of the pugilistic action, and the magnitude of the impact delivered (*Walilko, Viano & Bir, 2005*; *Kozin et al., 2021*).

However, no single athletic performance characteristic dominates in combat sports (*Slimani et al., 2016*). This demands that boxers maintain a certain level of speed, strength, endurance, agility, flexibility, and coordination throughout the competition (*Slimani et al., 2017*). These factors have been demonstrated to exhibit a direct correlation with athletic performance (Chaabène et al., 2015; *Loturco et al., 2016*). Punching is a rapid whole-body multi-planar muscular movement, such as jabs and crosses, which is a commonly used technique in boxing; additionally, upper- and lower-body muscular strength may affect the kinematic and kinetic features of jabs and crosses (*Dinu et al., 2020*). Specifically, athletes who possess superior strength, explosive power, coordination, and responsiveness typically execute high-quality technical maneuvers with greater efficacy. Consequently, an integrated physical fitness training regimen that encompasses these attributes is likely to augment the athletic performance of boxers significantly.

The physical training programs currently adopted by boxers tend to focus on one or a few limited aspects of physical fitness exercises rather than a comprehensive one. Strength (*Loturco et al., 2016*; *Dunn et al., 2022*), explosive power, aerobic capacity (*Davis, Leithäuser & Beneke, 2014*; *El-Ashker et al., 2018*), coordination (*Çetin et al., 2018*), and speed agility (*Sheppard et al., 2006*) should be considered key components when designing a physical training program for boxers. Therefore, boxers need a more comprehensive and efficient training method that can improve athletic performance in a shorter period. Integrated neuromuscular training (INT) combines many training components, including resistance, balance, core strength, dynamic stability, agility exercises, and plyometric training (*Myer & Faigenbaum, 2011*). The effectiveness, completeness, and efficiency of INT have led to its widespread adoption by numerous sports teams as an integral part of their physical training regimen. Specifically, INT has demonstrated the ability to enhance neuromuscular control and increase functional joint stability. It can concurrently impact the sensitivity and responsiveness of the central nervous system while enhancing athlete strength through targeted motor unit activation and coordination as well as increased muscle activation (*Fort-Vanmeerhaeghe et al., 2016*). The purpose of INT was initially to enhance athletic performance and reduce the risk of injuries in young athletes (*Sañudo et al., 2019*; *Faigenbaum et al., 2014*; *Filipa et al., 2010*). The principle is applicable to basketball, football, badminton, volleyball, and table tennis (*McLeod et al., 2009*; *Pasanen et al., 2009a*; *Sugimoto et al., 2014*; *Trajković & Bogataj, 2020*; *Zhao et al., 2021*; *Xiong et al., 2022*). At the present time, this training method has not been applied to boxing programs, and there are no studies demonstrating the effects of short-term INT on the athletic performance of elite female boxers in the Chinese national team.

The objective of this study was to investigate the effect of three weeks of INT on the athletic performance of female boxers, which will inform the development of training programmes for boxers.

## MATERIAL AND METHODS

### Participants

Thirty-seven female boxers (Age: 26.00 ± 3.11 years; mean height 171 ± 7 cm; mean weight 62.86 ± 9.08 kg; training experience 9.37 ± 2.31 years) from the Chinese women's boxing team who were preparing for the Paris Olympic Games volunteered to participate in the study. The time of study was January to February 2022. These athletes had all participated in national or international competitions and thus have a high level of competitive ability. All participants were familiarized with the study procedures before the start of the study and provided written informed consent after fully understanding the risks and benefits associated with it. The inclusion criteria for this study were as follows: (1) All had participated in the National or World Series and were selected in the national team; (2) at least five years of training experience; (3) at least 18 years old; (4) had no injury problems in the past six months and could participate in the training intervention; and (5) At least 95% of the training content was completed during the intervention period. Exclusion criteria were (1) athletes with muscle or joint injuries; (2) failed to participate in the training or quit halfway; (3) athletes who had not completed more than 95% of the training content. Given that the participants involved in this study were national team athletes, no other independent control groups of the same level could be paired for the comparison of the effects of concurrent training at the time of study design. Therefore, this experiment used the index test data before the intervention training as the baseline standard to compare and assess the magnitude of the response to the training intervention.

### Overview of procedures

All tests and interventions were performed at the Baisha Professional Boxing Training Base of the Chinese Boxing Association (Hainan, China). The baseline test consisted of three independent test days in which basic physical fitness and hitting ability measures were completed. On the first day, the vertical jump, 1-min hexagon jump and 3-min double shake were performed in the morning, and 30-m run and 400-m run were performed in the afternoon. On the second day, bench press (BP) and squat tests were performed in the morning, and 3,000-m running was performed in the afternoon. In the afternoon of the third day, a single punch test and a 10-s, 30-s, and 3-min continuous strike test were performed. Participants were asked to refrain from staying up late and drinking alcohol before taking the test. The post-training test was procedure-wise completely consistent with the baseline test. The study was approved by the Ethics Committee of the Shanghai University of Sport (code: 102772021RT029, approval date: 2021/3/12).

The training schedule is given in Table 1. This three-week training schedule only included the intervention of the INT protocol, and no skill and tactical exercises were performed. The design of INT program is based on the physical training needs of boxing (*Davis et al., 2015*; *Davis et al., 2016*; *Vasconcelos et al., 2020*; *Beattie & Ruddock, 2022*), incorporating

the results of previous studies (*Myer et al., 2005*; *Myer et al., 2006*; *Myer et al., 2011a*; *Myer et al., 2013*; *Faigenbaum & Myer, 2010*; *Emery et al., 2015*; *Fort-Vanmeerhaeghe et al., 2016*) and seeking specific comments and suggestions from coaches and experts. Strength, speed, plyometric training, core stability, and functional training were combined. The main types of training are resistance training, speed interval training, core stability training, and energy system metabolic training. In contrast to traditional training approaches that typically emphasize one or a few limited physical fitness aspects in previous studies, the INT of this study was based on strength training throughout the training protocol while focusing on speed, dynamic stability, agility, and proprioception. For high-intensity training schedules, such as resistance training, no more than three times per week, with functional training, core stability training was included as a transition between non-consecutive training days to allow adequate recovery time for athletes (*Faigenbaum et al., 2009*; *Myer & Faigenbaum, 2011*). INT combined with aerobic training was found to be more effective than INT alone in influencing the explosive power of the athletes; therefore, in the present study, no more than three aerobic training sessions per week were scheduled on top of INT (*Balaganesh Dharuman, 2022*). The participants in this study were elite-level athletes who had prior experience in high-level training. Therefore, 11 sessions per week were performed for a total of 33 sessions, each of which was followed by approximately 15 min of warm-up, followed by approximately 60 to 90 min of INT. Table 1 shows a 3-week INT daily type training. Details of the training are uploaded in the (Tables S1–S11).

## Assessment of physical fitness variables
### 1RM test
Maximum strength was assessed using the 1RM test for all participants (*Ryman Augustsson & Svantesson, 2013*). The test began with a 3-min light weight warm-up, after which participants began squats or bench presses at a predetermined weight. This weight was less than their 1RM, and the weight was increased according to the actual situation as reported by the participant. Participants were given a 2–3min rest period between two consecutive trials. The final weight at which participants could only repeat once was considered 1RM, and all participants reached their 1RM within 3–5 sets.

### Vertical jump test
Portable jump test system (Smart jump, Fusion Sport Inc., Australia) used to evaluate athletes' lower extremity explosiveness (*Maulder & Cronin, 2005*). The participants stood on the force board, and the indicator light was lit to start the test. After preparing the posture, the participants were tasked to fully swing their arms vertically and jump as high as possible. When the indicator light turned green for the next vertical jump test, bending the knees and hips was no longer allowed in the jumping process. Each participant had 3 test opportunities, with 30s rest between each test, and the best score was taken for data analysis. Prior to data collection, the research assistant guided the participants, who performed 3–5 attempts to familiarize themselves with the test. At least a 2-min rest period was set between the instruction test and the actual data collection.
**Table 1** **Overview of the 3-week INT daily type of training.**

| | | Monday | Tuesday | Wednesday | Thursday | Friday | Saturday |
|---|---|---|---|---|---|---|---|
| **First week** | AM | Aerobic training core stability | Plyometric training core stability | High intensity interval training | Agility training core stability | Plyometric training | High intensity interval training |
| | PM | Functional training 2 Core Stability | Functional training 2 Core Stability | Rest | Functional training 2 | Aerobic training Core Stability | Functional training 2 |
| **Second week** | AM | Aerobic training Core Stability | Plyometric training Core Stability | High Intensity Interval Training | Agility training Core Stability | Plyometric training | High Intensity Interval Training |
| | PM | Functional training 2 Core Stability | Functional training 2 Core Stability | Rest | Functional training 2 | Aerobic training Core Stability | Functional training 2 |
| **Third week** | AM | Plyometric training Core Stability | Aerobic training Core Stability | Explosive Power Training Core Stability | Functional training 3 Core Stability | Explosive Power Training Core Stability | Aerobic training Core Stability |
| | PM | Sprint Interval Training Functional training 3 | Functional training 3 | Sprint Interval Training | Rest | Functional training 3 Core Stability | Functional training 3 Core Stability |

**Notes.**

AM, ante meridiem; PM, post meridiem.

### 30 m sprint run test

Portable speed test system (Smart speed, Fusion Sport Inc., Australia) used to evaluate athletes' sprinting ability (*Castagna et al., 2018*). Using a straight track, we measured a distance of 30 m, marked the start and finish lines with a marker, and placed two photogates at the start and finish. Using a standing start, the subject stood in the designated position and then started to sprint through the first photoelectric gate on their own when they were ready, and the test ended when they passed through the second photoelectric gate. Two tests were performed for each person, with a rest of at least 2 min for each test, and the best score was recorded at accurately 0.01 s.

## 400 m and 3,000 m running tests

A stopwatch was used for recording of 400 m and 3,000 m running time (*Bosquet, Léger & Legros, 2002*; *Vilmi et al., 2016*). Athletes use a standing start, starting the timer after the start in the prescribed track to complete the entire run and not cross track. The athlete reached the finish line with the trunk, and the test score was recorded to an accuracy of 0.01 s.

### Agility test

Evaluation was performed by 1 min hex hop and 3 min double roll (*Wu et al., 2024*). (1) 1 min hexagon jump: A hexagon is marked on the floor, each side should be 24 in (60.5 cm) long, and each angle should be set at 120°. The athlete was asked to stand with both feet in the middle of the hexagon. Start the clock on the command. The athlete jumped clockwise to the hexagon and then quickly jumped back to the specified origin. The process was repeated until the test time stopped, and the number of jumps was recorded within 1 min. (2) 3 min double shake: Jump rope and stopwatch were used, and the subject's double shake test began after starting. If the athlete cannot complete a double roll, the single score will not be counted. The athlete was instructed to complete the double roll as quickly as possible in 3 min, and the number of completed times was recorded after the end of time.

### Punching ability test

Xingxun Boxing Training Monitoring System (Xingxun BX260 2.0, Shanghai, China; Version: 2.0) and sandbags were used to evaluate the single-punch ability and continuous punching ability of athletes (*Kamandulis et al., 2018*). When a single punch test is performed, the participant is prepared to punch according to her dominant side. On command, the test system is activated and the participant performs the test in a sequence of straight punch by lead hand, straight punch by rear hand, uppercuts punch by lead hand, uppercuts punch by rear hand, hooks punch by lead hand and hooks punch by rear hand. Five punches are performed for each punching technique, and the test requires that the punches be performed at full power. The punches were all executed at full power, with the best score being recorded after each of the five punches. After the completion of the single punch test, participants rested for 3 min. Then 10-second, 30-second, and 3-minute continuous punching was performed. Final record of punch power.
**Table 2 Indicators related to the basic physical fitness test.**

| | Test Indicator | Pre-test | Post-test | ES | Change/% |
|---|---|---|---|---|---|
| Strength | Bench press (kg) | 57.89 ± 7.65 | 66.91 ± 10.55 | 1.66 | 15.54 ± 8.80* ↑ |
| | Squat (kg) | 77.70 ± 7.94 | 100.97 ± 12.39 | 2.10 | 30.47 ± 15.2* ↑ |
| Explosiveness | Vertical jump (cm) | 32.23 ± 5.19 | 36.99 ± 5.82 | 2.12 | 15.04 ± 7.82* ↑ |
| Speed | 30 m sprint (s) | 4.80 ± 0.23 | 4.70 ± 0.20 | 1.19 | 2.14 ± 1.74* ↓ |
| Endurance | 400 m (min) | 1.20 ± 0.07 | 1.18 ± 0.07 | 0.42 | 1.62 ± 3.96* ↓ |
| | 3,000 m (min) | 13.09 ± 1.12 | 12.86 ± 0.84 | 0.34 | 1.52 ± 4.87* ↓ |
| Agility (Times) | 1-minute hexagonal jump | 64.75 ± 7.39 | 78.40 ± 6.25 | 2.36 | 22.01 ± 11.40* ↑ |
| | 3-minte double shake | 261.10 ± 41.08 | 302.16 ± 38.87 | 1.60 | 16.75 ± 11.89* ↑ |

**Notes.**
↑Indicates an increase.
↓a decrease compared with the pre-experiment.
gEffect size measured by Hedges' g.
*significant difference from baseline ($p < 0.05$).

## Statistical analysis

All statistical analyses were performed using SPSS (version 27 IBM, Armonk, NY, USA) for data processing, and results were presented as mean ± standard deviation. Shapiro–Wilks was used to test the normality of the data before statistical analysis, and all data were normally distributed. Given that this study had no comparison group, paired sample t test was used to analyze the significance of the data before and after the test to illustrate whether any difference existed in the data before and after the test. $P < 0.05$ indicated significant difference and was statistically significant. To evaluate the effect size, Hedges' g (effect size, ES) was calculated. To estimate within-group pairwise effect sizes (*Fritz, Morris & Richler, 2012*), ES was interpreted as tiny (ES ≤ 0.20), small (0.20 < ES ≤ 0.60), medium (0.60 <ES ≤ 1.20), large (1.20 < ES ≤ 2.00), and very large (ES ≥ 2.00).

## RESULTS

After the three-week INT intervention, all participants completed all training and testing. Comparative analysis with the pre-experimental baseline values showed a significant increase in all basic fitness test indicators of the athletes (Table 2) (Fig. 1). The 1RM of BP increased by 9.02 kg (16.95 ± 9.27%, $p < 0.05$), and the 1RM of squat increased by 23.27 kg (30.47 ± 15.2%, $p < 0.001$). The vertical jump was significantly improved by 4.76 cm (15.4 ± 7.7%, $p < 0.001$). The athlete's 30 m sprint running time was significantly increased by 0.1 s (2.14 ± 1.74%, $p < 0.01$). Significantly higher test scores in 400 m run (1.62 ± 3.96%, $p < 0.05$) and 3,000 m run (1.52 ± 4.87%, $p < 0.05$). The number of hexagonal jump at 1 min and double shake at 3 min increased by 13.65 (22.01 ± 11.40%, $p < 0.05$) and 401.06 (16.75 ± 11.89%, $p < 0.05$), respectively.

Athletes' 10-second punching, 30-second punching, and 3-minute punching test scores were significantly improved compared with baseline data (Table 3) (Fig. 1). The 10-second punching increased by 184.84 kw (36.32 ± 32.53%, $p < 0.05$), the 30-second punching increased by 348.89 kw (27.16 ± 29.39%, $p < 0.001$), and the 3-minute punching increased

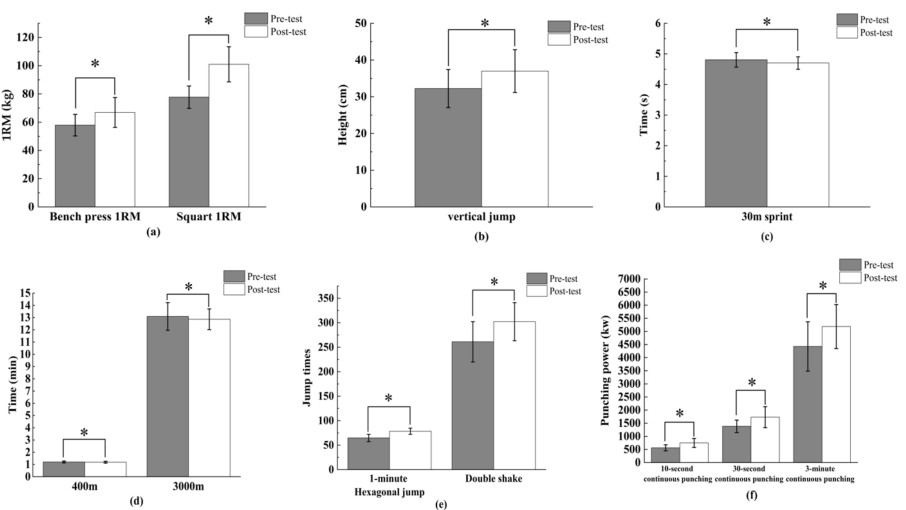

**Figure 1** (A) 1RM test of the bench press and squat pre-and post-INT; (B) Vertical jump test pre-and post-INT; (C) 30 m sprint run test pre-and post-INT; (D) 400 m and 3,000 m run tests pre-and post-INT.

**Table 3 Indicators related to specialized punching ability test.**

| Test indicator (kw) | | Pre-test | Post-test | ES | Change/% |
|---|---|---|---|---|---|
| Straight punch | Lead | 17.08 ± 3.22 | 18.39 ± 4.77 | – | 11.39 ± 36.24 ↑ |
| | Rear | 20.84 ± 5.20 | 22.52 ± 4.59 | – | 13.59 ± 31.96 ↑ |
| Hooks punch | Lead | 18.69 ± 5.20 | 20.15 ± 5.92 | – | 17.18 ± 52.84 ↑ |
| | Rear | 28.18 ± 7.44 | 30.18 ± 10.13 | – | 12.86 ± 45.14 ↑ |
| Uppercuts punch | Lead | 17.08 ± 3.22 | 18.39 ± 4.77 | – | 11.39 ± 36.24 ↑ |
| | Rear | 21.01 ± 9.66 | 23.03 ± 7.26 | – | 30.90 ± 70.41 ↑ |
| Continuous punching | 10-second continuous punching | 562.53 ± 114.87 | 747.37 ± 170.21 | 1.07 | 36.32 ± 32.53* ↑ |
| | 30-second continuous punching | 1,381.40 ± 237.91 | 1,730.29 ± 399.01 | 0.87 | 27.16 ± 29.39* ↑ |
| | 3-minute continuous punching | 4,426.23 ± 940.81 | 5,185.43 ± 839.20 | 1.01 | 19.80 ± 20.56* ↑ |

**Notes.**
*Significant difference from baseline ($p < 0.05$).

by 759.20 kw (25.80 ± 28.77%, $p < 0.05$). However, there was no significant difference existed in the test results of single-punch ($p > 0.05$).

## DISCUSSION

This study is the first to examine the effects of INT on the athletic performance of elite female boxers. Some experimental results support the hypothesis of this study. Specifically, INT can significantly improve an athlete's strength (1RM), explosiveness (vertical jump), sprinting ability (30 m run), energy system metabolism (400 m and 3,000 m runs), agility (hexagonal jumps and double cranks), and continuous punching performance

(10-second, 30-second, and 3-minute continuous punching). However, the ability to land a single-punch was not improved significantly.

The strength training protocol for this study was designed on the basis of the 1RM of the athletes. Studies have found that high load strength training (3–6RM) is associated with maximal force production but not muscle hypertrophy (*Docherty & Sporer, 2001*).

In contrast to previous studies, INT protocol used in this study was a diverse range of muscle strength training. The use of general strength exercises (*i.e.,* BP, squat, *etc.*) during the first week of training is intended to utilize high-intensity training to produce greater mechanical stimulation of the neuromuscular system, which in turn has a favorable effect on the athlete's maximal strength (*Izquierdo et al., 2002*). During the second to third weeks, plyometric training (*e.g.,* high pull + jump box, squat + vertical jump with elastic band) is performed to convert maximum force into elastic potential energy. This training is characterized by a forceful and rapid completion of movements, a high output of force, and reduced foot contact with the ground (*Wang et al., 2022*; *Kim & Lee, 2023*). After three weeks of INT intervention, BP and squat of athletes increased by 15.54% and 30.47%, respectively. The results of the present study suggest that INT is effective in improving strength performance in athletes, which is consistent with other studies (*Fatouros et al., 2000*; *Li et al., 2019*; *Panagoulis et al., 2020*). *Faigenbaum et al. (2014)* found that the 1RM squat test in the INT group improved compared with the control group. A recent study of an 8-week INT intervention in elite female table tennis players showed an 11.6% increase in 1RM squats in the INT group after training (*Xiong et al., 2022*). Increased strength and explosiveness are critical to a boxer's athletic performance. An analysis of the kinematics of straight right punches in eight elite amateur boxers found that the amateur boxers attempted to target their opponent from longer distances and emphasize high punch speed, while the professional boxers attempted to deliver heavier punches from closer distances (*Cheraghi et al., 2014*). Therefore, athletes of different levels have obvious differences in technical movements such as punching speed and accuracy, and athletes with better power and explosive power tend to complete technical movements with higher quality.

Regarding lower limb motor performance, such as vertical long jumps, a relationship exists between both muscle strength and power (*Nuzzo et al., 2008*; *Alemdaroğlu, 2012*). After the INT intervention, vertical jump height increased by 15.04%, possibly due to improved synchronization of motor units, increased efficiency of the stretch-shortening cycle (SSC), and enhanced coordination among body parts (*Fatouros et al., 2000*). This finding is consistent with previous research (*Chaouachi et al., 2014*; *Panagoulis et al., 2020*; *Xiong et al., 2022*). The main reason for the difference in experimental results between the different studies may be related to the age, gender, and motor basis of the subjects. Weight-bearing exercises combined with plyometric training were found to improve vertical jumping and increase leg muscle strength (*Fatouros et al., 2000*). Thus, the strength training schedule of INT protocol of this study may have resulted in an improvement in vertical jump performance. In addition, many variants of jump practice are available in the INT protocol that also have an additional impact on the training effect of athletes. However, the measures tested in this study were too large to demonstrate the benefits

beyond those measured, and further research is needed to determine the extent to which INT is linked to performance in elite athletes. It can be said that, the increase in vertical jump height of athletes is most likely the result of the development of a comprehensive training program.

Most studies have demonstrated significant improvements in athletic performance with INT or speed and agility training interventions only 2–3 times per week (*Miller et al., 2006*; *Mujika, Santisteban & Castagna, 2009*; *Pasanen et al., 2009b*; *DiStefano et al., 2010*; *Buchheit et al., 2010*; *O'Malley et al., 2017*). The study revealed significant improvements ($p < 0.001$) in athletes' performance on the 30 m sprint (2.14%), 1 min hexagonal jump (22.01%), and 3 min double shake (16.75%) tests after the 3-week INT intervention, supporting our previous hypothesis. The results of a 5-week study of INT in tennis players indicated different levels of improvement in 5 m (2.4%), 10 m (1.4%), and 20 m (4.9%) sprint performance, respectively (*Fernandez-Fernandez et al., 2018*). Another study with netball athletes also found significant improvements in post INT 20 m sprint times, 505 test times at six weeks (*Hopper et al., 2017*). A study (*Anbu, Ss & Dharuman, 2022*) showed that the agility performance of the neuromuscular training before soccer game practice and the neuromuscular training after soccer game practice groups improved by 12.57% and 8.75%, respectively. In similar findings, the use of plyometrics in INT was found to have a significant effect on one-legged jumping ability (*Myer et al., 2006*; *Faigenbaum et al., 2014*). Notably, the reason for the design of the sprinting and agility training in this study is that boxing involves high-speed movement and dodging in different directions in a limited space, and athletes rarely have the opportunity to reach their maximum speed. Thus, initial acceleration is a key component of athletic performance. On the one hand, performance improves because the increase in 1RM positively affects speed, and a significant correlation exists between maximum power and short sprint time (*Haff & Nimphius, 2012*). On the other hand, the functional training in INT program of this study was designed with many movements similar to those specific to boxing, including frequent exercises on quick steps and fast movement abilities, which improve the athlete's reflexes and initiation abilities (*Arseneau, Mekary & Léger, 2011*). In addition, INT has been shown to affect the sensitivity and responsiveness of the central nervous system and to improve strength in athletes through coordination and control of motor units (*Fort-Vanmeerhaeghe et al., 2016*). Therefore, improvements in athletes' sprinting ability and agility may be related to improvements in lower limb muscle strength and neural components (*e.g.*, coordination between lower limb muscles, stride frequency) (*Brughelli et al., 2008*).

After 3 weeks of INT intervention, the 400 m and 3,000 m test scores of athletes were significantly improved ($p < 0.05$). This finding is consistent with our previous research hypothesis that INT improves fatigue resistance in elite female boxers (*Faigenbaum et al., 2014*; *Fernandez-Fernandez et al., 2015*; *Fort-Vanmeerhaeghe et al., 2016*; *Akbar et al., 2022*). By studying the effects of the two INT methods on the athletic performance of athletes, INT combined with aerobic training was found to have a more significant effect on the athletic performance of athletes than INT alone, and our study also proved this point (*Balaganesh Dharuman, 2022*). Improving a boxer's resistance to fatigue is critical; boxers need to maintain accuracy and power of their strikes, strength of movement, and

defensive maneuvers to win matches in a state of exhaustion, which requires strong aerobic and anaerobic capacity (*Arseneau, Mekary & Léger, 2011*; *Davis, Wittekind & Beneke, 2013*; *Bruzas et al., 2014*).

The improvement in performance is related to the design of INT protocol of the present study. In the present study, aerobic exercises, multiple repetitions with short intervals, such as high-intensity interval training and repetitive sprint training, with heart rate intervals of 160 to 190 bpm were employed during training higher intensities and shorter intervals may burden both aerobic and anaerobic metabolic systems, resulting in higher neuromuscular loads (*Tabata et al., 1996*; *Buchheit & Laursen, 2013*; *Kamandulis et al., 2018*; *Vasconcelos et al., 2020*). This process is sufficiently effective to improve adaptations for endurance performance (*Laursen & Jenkins, 2002*; *Kamandulis et al., 2018*). Specifically, repetitive short-distance exercise not only stresses many of the physiological and biochemical systems used in aerobic exercise (*Gibala et al., 2006*; *Burgomaster et al., 2008*) but also induces changes in glycolytic enzymes, muscle buffering, and ionic regulation that improve anaerobic capacity (*Harmer et al., 2000*; *Bishop et al., 2009*; *Dawson, 2012*).

In the present study, after three weeks of INT intervention, the continuous punching ability of athletes significantly improved, but no significant change was noted in single punch ability. This situation may be related to the arrangement of the training program. First, previous studies have mostly placed INT before or after specialized training to play a role in specialized warm-up or injury prevention (*Steffen et al., 2013*; *Panagoulis et al., 2020*; *Anbu, Ss & Dharuman, 2022*). This study used INT as a special physical fitness training intervention, which did not involve the practice of specific special techniques. The latter may cause a decline in athletes' hitting ability during the test. Another possible plausible explanation relates to the residual fatigue effects of excessive exercise in athletes, in which the nervous, muscular, and metabolic systems of athletes are not fully recovered (*Fatouros et al., 2000*; *Sáez de Villarreal et al., 2013*). Therefore, athletes may conceivably make substantial progress if the number of exercises is reduced to 2–3 times per week and the duration of the intervention is extended. More research is needed to prove this inference.

In a boxing match, athletes must continuously throw punches, and the force and velocity of each punch can determine the outcome of the match (*Bianco et al., 2013*; *Loturco et al., 2016*). The ability to perform 10s, 30s and 3 min continuous punching is a combination of the athlete's strength, speed agility, explosiveness, and metabolic capacity of the energy system. Athletes in this study averaged 36.32%, 27.16%, and 19.80% increases in 10-second, 30-second, and 3-minute continuous punching after the 3-week INT intervention, respectively. Regardless of weight category, boxers who achieved higher cumulative power (number of strikes multiplied by the impact generated by each punch in the match) and more punches won the fight (*Pierce et al., 2007*). In combat sports, generating explosive force in the upper body is necessary to deliver powerful punches to achieve the desired athletic performance (*Loturco et al., 2018*). One study examined the relationship between BP power and punch performance and found that 80% of 1RM BP exercises were significantly correlated with punch performance (*López-Laval et al., 2020*). While the arms transmit power at the end of the power chain, the importance of

lower body strength should not be overlooked. For example, biomechanical analyses of the "straight punch" show a 15% increase in punching power due to increased lower body strength that enhances upper body kinetic chain movement (*Tillin et al., 2010*). The increase in continuous punching capacity of the athletes in this study may be related to the base strength and explosive endurance that promotes muscle growth in the INT program, the elevated metabolic capacity of the energy system, and the functional training required for specialization. It reflects that the improvement of athletes' basic physical fitness has a positive role in promoting the performance of special sports. In addition, improved continuous punching ability may be associated with Plyometric training reduced time to complete the stretch-shortening cycle, increased cross-sectional area of fast muscle fibers, improved neural activation, changes in intrinsic muscle properties, increased myosin-ATP activity, and faster synchronization of motor units or higher firing frequency (*Gorostiaga et al., 2006*). However, the present study unable to demonstrate the intrinsic variability described above, which requires further research to argue for an intrinsic link between INT and sport-specific performance. In conclusion, the improvement of a boxer's striking ability is closely related to the development of the body's general ability, and INT seems to be more efficient and more in line with the physical demands of boxing than training with only one method in isolation.

There are some limitations in this study: (i) the lack of assessment of risk factors for injury and the lack of assessment of neuromuscular and muscle morphological changes, (ii) control group could not be set up in this study due to the high exercise level of the subjects and the limitation of the actual experiment. In addition, the improvement in motor performance in this study was largely attributed to neuromuscular adaptation. For a better understanding on the underlying physiological mechanisms of INT-related adaptation, a longer duration of intervention with simultaneous assessment of nervous system and muscle adaptation using EMG, ultrasound, or other imaging techniques is recommended.

## CONCLUSION

The results of the study showed that a 3-week INT intervention significantly improved the athletic performance of female boxers. Improvements in performance on the deep squat, BP, vertical long jump, 30 m sprint, 400 m and 3,000 m run, 1min hexagonal jump, and 3 min double shake and continuous punching ability tests demonstrated the relevance of INT as a comprehensive training modality for elite boxers. INT can be used as a routine physical training method to improve the performance of boxers and promote the improvement of sports performance.

### Funding

This study was funded by the "The General Administration of Sport of China Preparation for the Paris Olympic Games National Boxing Team Key Athletes Key Strength Ability Enhancement Service Work (Project No. 2023AY011)". Science and Technology Commission of Shanghai Municipality Program Project: "Research on Key Technologies for Improving the Athletic Performance of Combat Sports in the Paris Olympics through Scientific and Technological Assistance" (Project No. 22010503800). The funders had no role in study design, data collection and analysis, decision to publish, or preparation of the manuscript.

### Grant Disclosures

The following grant information was disclosed by the authors:
The General Administration of Sport of China Preparation for the Paris Olympic Games National Boxing Team Key Athletes Key Strength Ability Enhancement Service Work: 2023AY011.
Science and Technology Commission of Shanghai Municipality Program Project: "Research on Key Technologies for Improving the Athletic Performance of Combat Sports in the Paris Olympics through Scientific and Technological Assistance": 22010503800.

### Competing Interests

The authors declare there are no competing interests.

### Author Contributions

- Zhen Niu conceived and designed the experiments, performed the experiments, analyzed the data, prepared figures and/or tables, authored or reviewed drafts of the article, and approved the final draft.
- Zijing Huang analyzed the data, authored or reviewed drafts of the article, and approved the final draft.
- Gan Zhao analyzed the data, prepared figures and/or tables, and approved the final draft.
- Chao Chen conceived and designed the experiments, performed the experiments, analyzed the data, prepared figures and/or tables, authored or reviewed drafts of the article, and approved the final draft.

### Ethics

The following information was supplied relating to ethical approvals (i.e., approving body and any reference numbers):
The Ethical Committee of the Shanghai University of Sport granted approval to conduct the study (Ethical Application Ref: 102772021RT029).

### Data Availability

The raw data are available in the Supplemental File.

## Supplemental Information

Supplemental information for this article can be found online at http://dx.doi.org/10.7717/peerj.18311#supplemental-information.

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
