# Peer review of "Impact of three weeks of integrative neuromuscular training on the athletic performance of elite female boxers"

_PeerJ, doi:10.7717/peerj.18311_

## Round 0.1 · original submission · Minor Revisions

Based on the reviewers' comments, I recommend a careful revision of the manuscript, especially considering that the results are interesting.

·

Basic reporting

The English language need to improve.

Introduction and background to show context. Literature well referenced and relevant.

Structure conforms to PeerJ standards, discipline norm.

Figures are relevant, well labeled and described. The image size is very small and creates difficulty of reading. Please make it clear and readable.

Raw data has been supplied.


Section wise Review

Title: Informative

Author Affiliation
Page No: 05, Line no: 08:College of Physical Education

Abstract
Page No: 06, Line no: 20: Please mention the age and level of participation of the athletes.

Page No: 06, Line no: 28: write (p<0.05)

Page No: 06, Line no: 29: write (p<0.05)

Page No: 06, Line no: 30: write (p<0.05)

Page No: 06, Line no: 31-33: The 3-week INT can significantly improve the maximum strength, vertical explosive power, linear acceleration, agility, and singular and continuous punching abilities of Chinese elite female boxers.

Page No: 06, Line no: 35: Key words must be more specific.

Introduction
Page No: 07, Line no: 42: Amateur boxing ……. .. good physical fitness and physiological health …………..

Page No: 08, Line no: 76-77: Female boxers have a tendency to have a higher body fat percentage, better aerobic capacity, and less explosive power than male boxers (Chaabène et al., 2015).

Page No: 08, Line no: 81-83: Therefore we hypothesized that 3-week INT would improve athletic performance in elite female boxers. We aim to explore the effect of 3-week INT training on the performance of female boxers and subsequently provide reference for the formulation of training plan for boxers .

Materials and Methods
Participants
Page No: 08, Line no: 86: Please use cm for unit of height.

Page No: 08, Line no: 88: Please mention the time of study.

Page No: 08, Line no: 88-89: The characteristics of the participants are not shown in Table 1.

Overview of procedures
Page No: 09, Line no: 117-118: A year-long cycle of training from January 2022-February 2023 in preparation for the March 2023 International Boxing Federation Women’s Boxing World Championships.

Page No: 09, Line no: 119-120: Write. The training schedule is given in Table....

Page No: 09, Line no: 119-120: Please use one table to show the three week training schedule.

Page No: 09, Line no: 121: The design of our ….

Page No: 09, Line no: 140:Tables 1, 2 and 3 show the different training modules for each 141 day of the week with the specific training content shown in the supplemental information.

Page No: 09, Line no: 143: Testing protocols

1RM test, Vertical jump test, 30m sprint run test, 400m and 3000m running tests, Agility test. Punching ability test.

Page No: 10, Line no: 167: We used a stopwatch for the 400m track and field .

Statistical Analysis
Page No: 10, Line no: 198: …..paired sample t test…

Page No: 11, Line no: 203: Please give references for Statistical analysis.

Result
Page No: 11, Line no: 205: Please write ‘Results’.

Discussion
Page No: 11, Line no: 223: To our knowledge ,…

Page No: 11, Line no: 231-232: In contrast to previous studies, our INT protocol used a diverse range of muscle strength training.

Page No: 12, Line no: 240: Our data suggest that INT is effective in improving strength performance in athletes, which is consistent 242 with other studies (Fatouros et al., 2000; Li et al., 2019; Panagoulis et al., 2020).

Page No: 12, Line no: 242: For example ,….

Page No: 12, Line no: 262-263: Thus, the strength training schedule of our INT protocol may have resulted in an improvement in vertical jump performance.

Page No: 12, Line no: 267-268: In conclusion, the increase in vertical jump height of athletes is most likely the result of the development of a comprehensive training program.

Page No: 12, Line no: 288-291: On the other hand, the functional training in our INT program was designed with many movements similar to those specific to boxing, including frequent exercises on quick steps and fast movement abilities, which improve the athlete’s reflexes and initiation abilities (Arseneau, Mekary & Léger, 2011).

Page No: 13, Line no: 289-300: This finding is consistent with our previous research hypothesis that INT improves fatigue resistance in elite female boxers (Faigenbaum et al., 2014a; Fernandez-Fernandez et al., 2015; Fort-Vanmeerhaeghe et al., 2016b; Akbar et al., 2022).

Page No: 13, Line no: 307-308: The improvement in performance is related to the design of our INT protocol.

Page No: 13, Line no: 308-310: First, we included aerobic exercises. Second, we included multiple repetitions with short intervals, such as high-intensity interval training and repetitive sprint training, with heart rate intervals of 160 to 190 during training loads.

Page No: 13-14, Line no: 319-320: In our study, after 3 weeks of INT intervention, the continuous punching ability of athletes could be significantly improved, but the single punch ability could not be significantly improved.

Page No: 14, Line no: 331: However, research is needed to prove this inference.

Page No: 14, Line no: 347-350: The increase in continuous punching capacity of the athletes in our study may be related to the base strength and explosive endurance that promotes muscle growth in the INT program, the elevated metabolic capacity of the energy system, and the functional training required for specialization.

Page No: 14, Line no: 356-357: However, we were unable to demonstrate the intrinsic variability described above, which requires further research to argue for an intrinsic link between INT and sport-specific performance.

Page No: 15, Line no: 361-364: This study has some limitations. The first is the lack of assessment of risk factors for injury and the lack of assessment of neuromuscular and muscle morphological changes. Second, a control group could not be set up in this study due to the high exercise level of the subjects and the limitation of the actual experiment.

Page No: 15, Line no: 364-365: In addition, the improvement in motor performance in our study was largely attributed to neuromuscular adaptation.

Conclusion
Well written

Acknowledgments
Page No: 15, Line no: 379-381: The authors are thankful to the financial support received for this study from ……. (Ref. No). The authors are also thankful to the athletes, coaches, officials and healthcare providers for their participation in this study.

Reference
Please write ‘References’

Experimental design

The present work is original primary research within Scope of the journal.

Research question has been well defined, relevant and meaningful. It is stated how the research fills an identified knowledge gap.

Rigorous investigation has been performed to a high technical and ethical standard.

Methods described with sufficient detail and information to replicate. The authors are requested to provide reference for each physical fitness test performed.

Experimental Design was properly stated.

Please use one table to show the three week training schedule.

Validity of the findings

Impact of INT training has been assessed, however the training load and the risk of overload has not been assessed. Meaningful replication has been encouraged and benefit to literature is clearly stated.

The characteristics of the participants have not shown in Table 1. All other underlying data have been provided; they are statistically sound.

Conclusions are well stated, linked to original research question and limited to supporting results.

The research work has been conducted with an accurate method, with the right tools and conditions to yield acceptable and reliable data.

Additional comments

Verify correct reporting of raw data, which representing the performance of the athletes.
The image size is very small and creates difficulty of reading. Please make it clear and readable.
The research findings are unique and may be helpful to the athletes.
The training load and risk of over load training was not included in this study. This is one the limitation of this study.

There are some errors in the text which require minor revision.

Recommend Revision and Resubmission
May be accepted after proper Revision as suggested by the Reviewer.

·

Basic reporting

English is fine. References are fine. The manuscript structure is fine, and there is the raw data. The manuscript message is self-contained.

Experimental design

Research fulfils journal aims and scope. The research question is proper. The investigation was performed rigorously. Methods are well described*.
* missing control group

Validity of the findings

The findings are valid.

Additional comments

General comments
This manuscript aims to investigate the effects of integrative neuromuscular training (INT) on the athletic performance of elite female boxers and subsequently provide a reference for formulating training plans for boxers. The aims are commendable. Authors found that the INT can significantly improve maximum strength, vertical explosive power, linear acceleration, agility and singular and continuous punching abilities of boxers. I am not fully convinced whether it is acceptable or not not to have a control group. Maybe comparing experimental group results with a previous year’s cohort, which trained standard? Overall, the authors manage to fulfil their aims sufficiently.

Specific comments
Please justify the lack of a control group consistently.
(line 20) Some basic info on integrative neuromuscular training should be provided in the abstract.

Minor comments
(l48) … 2016). This…
(l612÷4, 424-5 and 409-10) please, amend citations;
(l141) in which specific supplemental information file?
(l151 and 160) model(s) and place?
(l167) 3000m running test implies 7.5 400m laps: two different timing points?
(182-3) place and country?
(l282) … of plyometrics in…

---

## Round 0.2 · accepted · Accept

The authors have addressed properly all the points raised by the two reviewers. The manuscript is ready for publication.

·

Basic reporting

Improvement in has been made in English language.
All the quarries have been made carefully.

Experimental design

The present work is original primary research within Scope of the journal.

Validity of the findings

The research work has been conducted with an accurate method, with the right tools and conditions to yield acceptable and reliable data.

Additional comments

Verify correct reporting of raw data, which representing the performance of the athletes.

·

Basic reporting

General comments
I do not have any further concerns about the manuscript. The authors addressed all the points raised by the two reviewers properly.

Experimental design

no comment

Validity of the findings

no comment

Additional comments

no comment